# Recovery of Post-Stroke Spatial Memory and Thalamocortical Connectivity Following Novel Glycomimetic and rhBDNF Treatment

**DOI:** 10.3390/ijms23094817

**Published:** 2022-04-27

**Authors:** Josh Houlton, Olga V. Zubkova, Andrew N. Clarkson

**Affiliations:** 1Department of Anatomy, Brain Health Research Centre and Brain Research New Zealand, University of Otago, Dunedin 9054, New Zealand; josh.houlton@otago.ac.nz; 2The Ferrier Research Institute, Gracefield Research Centre, Victoria University of Wellington, 69 Gracefield Road, Lower Hutt 5040, New Zealand; olga.zubkova@vuw.ac.nz

**Keywords:** prefrontal cortex stroke, cognitive impairment, GAGs, astrogliosis, growth factors, BDNF

## Abstract

Stroke-induced cognitive impairments remain of significant concern, with very few treatment options available. The involvement of glycosaminoglycans in neuroregenerative processes is becoming better understood and recent advancements in technology have allowed for cost-effective synthesis of novel glycomimetics. The current study evaluated the therapeutic potential of two novel glycomimetics, compound A and G, when administered systemically five-days post-photothrombotic stroke to the PFC. As glycosaminoglycans are thought to facilitate growth factor function, we also investigated the combination of our glycomimetics with intracerebral, recombinant human brain-derived neurotrophic factor (rhBDNF). C56BL/6J mice received sham or stroke surgery and experimental treatment (day-5), before undergoing the object location recognition task (OLRT). Four-weeks post-surgery, animals received prelimbic injections of the retrograde tracer cholera toxin B (CTB), before tissue was collected for quantification of thalamo-PFC connectivity and reactive astrogliosis. Compound A or G treatment alone modulated a degree of reactive astrogliosis yet did not influence spatial memory performance. Contrastingly, compound G+rhBDNF treatment significantly improved spatial memory, dampened reactive astrogliosis and limited stroke-induced loss of connectivity between the PFC and midline thalamus. As rhBDNF treatment had negligible effects, these findings support compound A acted synergistically to enhance rhBDNF to restrict secondary degeneration and facilitate functional recovery after PFC stroke.

## 1. Introduction

Stroke remains a leading cause of death and disability, worldwide [1,2,3], with cognitive impairments being reported in over a third of stroke patients [4]. Despite this, cognitive impairments remain largely understudied and poorly understood, with specific approaches for rehabilitation of cognitive function remaining a significant gap in stroke research [5]. In the hope of building our understanding of post-stroke cognitive impairments and to explore the underlying neurological mechanisms, we utilised a photothrombotic model of stroke targeting the prefrontal cortex (PFC) that presents with impaired spatial working memory, cognitive flexibility and altered PFC-hippocampal coherence [6,7,8,9]. More recently, we have demonstrated a stroke-induced loss of connections between the PFC and midline thalamic nuclei [8], connections that are critical for maintaining various cognitive domains including spatial working memory [9,10,11]. Furthermore, we found persistent reactive astrogliosis throughout grey and white matter regions surrounding the stroke site, reflecting causal mechanisms linked to secondary degeneration that may contribute to this loss in thalamocortical connections and/or onset of cognitive impairments [8] similar to what has also been observed in human post-mortem brains [12]. With this in mind, therapeutic interventions that can modulate reactive astrogliosis and either prevent or rescue a loss in thalamocortical connectivity may assist in recovery of cognitive function following stroke.

Although once thought to only contribute to the structural properties of the extracellular matrix (ECM), it is now well established that glycosaminoglycans (GAGs) actively govern a range of complex cell signalling pathways in the developing and adult brain [13,14]. For instance, modulation of GAG composition within perineuronal nets (PNNs) is thought to alter their permissiveness to axonal growth and synaptic plasticity in the adult brain [15,16,17]. Furthermore, several GAGs have been implicated in bi-directionally controlling growth of neuronal processes within peri-infarct regions beside the glial scar following preclinical stroke (i.e., chondroitin-sulphate inhibits, whilst heparan sulphate stimulates axonal growth) [18,19,20,21]. The use of HS-containing proteoglycans (HSPGs) such as glypican have also been shown to significantly reduce the thickness of the glial scar, enhance neurite extension and promote behavioural recovery, when administered one-week after middle cerebral artery occlusion (MCAo) in rats [20]. Despite yielding such properties, the therapeutic application of GAGs and glycomimetics remain sparse, largely due to the extreme difficulty and high costs that have been historically associated with synthesis and production of these substances [22,23]. Excitingly, advancements in technology have led to innovative synthetic and chemoenzymatic approaches that allow for cost-effective production of small chain GAG libraries [24,25,26].

Interestingly, many growth factors (GFs) are known to require the presence of extracellular GAGs to exert their biological function [27,28]. Indeed, GAGs themselves have been observed to act as medium-affinity co-receptors to help bind GFs (e.g., brain-derived neurotrophic factor, BDNF) to their respective receptors [29]. It has also been postulated that a combinational intervention may be required to maximise the chances of successfully translating a pharmacological treatment for stroke, from bench-to-bedside. Therefore, the current study sought to investigate the therapeutic effects of two novel glycomimetics, compound A and compound G, with or without rhBDNF, a GF crucial for functional recovery following stroke [30,31,32].

## 2. Results

### 2.1. Infarction Volume Quantification

On day-35 post-stroke, brains were collected and infarct volumes quantified using CV staining (Figure 1). A two-way ANOVA revealed no overall differences in hemispheric damage (*F*_1,124_ = 0.6183, *p* = 0.4332), with the damage to the left and right hemispheres following a bilateral PFC stroke being similar across all experimental groups. We did observe a significant overall effect of treatment on infarct volumes (*F*_6,124_ = 2.939, *p* = 0.0102); however, when adjusted for post-hoc comparisons, no differences were observed between any experimental groups, nor was an interaction effect identified. Furthermore, a one-way ANOVA failed to identify a significant effect of treatment on the average total stroke volume (*F*_6,62_ = 1.274, *p* = 0.2822), with total stroke volumes being similar across all stroke animals and compared to what has been published previously [6,7].

### 2.2. Effect of Compound A Treatments on Post-Stroke Spatial Memory

The therapeutic potential of our novel glycomimetics were assessed by comparing the performance of sham and stroke animals on OLRT (Figure 2 and Figure 3). Consistent with what has been previously reported [6], a two-way ANOVA assessing the exploratory preference of vehicle- and compound A-treated animals at one-week post-stroke failed to identify any treatment, stroke or interaction effects (Figure 2A). At four-weeks post-stroke, however, a significant overall effect of stroke only was observed (*F*_1,33_ = 38.70, *p* < 0.0001; Figure 2B). Post-hoc comparisons further revealed that vehicle- (49.58 ± 7.45%; *p* = 0.0118) and compound A-treated stroke animals (45.78 ± 13.86%; *p* < 0.0001) both spent significantly less time investigating the object in the novel location, relative to saline-treated sham controls (saline: 67.51 ± 5.82%) at this time.

Regardless of treatment (vehicle, compound A, rhBDNF or compound A + rhBDNF), a one-way ANOVA failed to report any significant overall treatment effects at both one- (Figure 2A) or four-weeks post-stroke (Figure 2B). These findings indicate that all stroke animals exhibited a similar delayed-onset impairment in ORLT performance at four-weeks post-stroke.

### 2.3. Effect of Compound G Treatments on Post-Stroke Spatial Memory

Consistent with what was reported in compound A datasets above, a two-way ANOVA assessing OLRT performance of vehicle- and compound G-treated animals at one-week post-stroke failed to show any significant stroke, compound G or interaction effects (Figure 3A). However, at four-weeks post-stroke, a significant overall effect of stroke only (*F*_1,33_ = 18.50, *p* = 0.0002) was reported on OLRT performance (Figure 3B). Post-hoc comparisons further reported that stroke animals, regardless of vehicle (49.58 ± 7.45; *p* = 0.0487) or compound G treatment (43.42 ± 14.09%; *p* = 0.0154), spent significantly less time exploring the object in the novel location relative to saline-treated shams (67.51 ± 5.82).

When assessing the combination of rhBDNF with compound G, a one-way ANOVA failed to report a significant effect of treatment at one-week post-stroke. In contrast, a significant overall treatment effect was identified at four-weeks post-stroke (*F*_3,33_ = 3.941, *p* = 0.0165; Figure 3B), with posthoc comparisons further confirming that stroke animals treated with compound G + rhBDNF (65.35 ± 7.04) spent significantly more time exploring the object in the novel location, relative to both vehicle- (*p* = 0.0341) and compound G-treated stroke animals (*p* = 0.0235).

### 2.4. Effect of Compound A Treatments on Thalamocortical Connectivity

Previous work in our laboratory has demonstrated a stroke-induced loss of connectivity between the PFC and several thalamic nuclei (nucleus reuniens, RE; mediodorsal thalamus, MD) {Houlton, 2020 #2300;Hillman, 2019 #1270}. To assess whether our glycomimetic compounds influenced changes in thalamocortical connectivity at this time, CTB+ve cell densities were calculated following prelimbic injections of CTB. A two-way ANOVA of vehicle- and compound A-treated animals reported an overall stroke effect, but no treatment or interaction effects, on the density of CTB+ve cells (cells/mm^3^) in the RE (*F*_1,32_ = 101.7, *p* < 0.0001; Figure 4A) and MD thalamic nuclei (*F*_1,32_ = 66.64, *p* < 0.0001; Figure 4C). Namely, post-hoc comparisons confirmed stroke-induced reductions in CTB+ve cell densities within the RE of vehicle- (1068 ± 176; *p* < 0.0001) and compound A-treated stroke animals (966 ± 103; *p* < 0.0001), compared to vehicle-treated sham controls (1596 ± 109). Moreover, a similar stroke-induced reduction in CTB+ve cell densities was observed in the MD of vehicle- (895 ± 74; *p* < 0.0001) and compound A-treated stroke animals (993 ± 168; *p* < 0.0001), compared to vehicle-treated shams (1235 ± 78).

Unexpectedly, a two-way ANOVA identified an overall effect of compound A treatment (*F*_1,32_ = 5.207, *p* = 0.0293), but not stroke or an interaction between these two factors, on the volume of the RE (Figure 4A). However, when adjusted for post-hoc comparisons this effect was lost, with sham animals displaying similar RE volumes irrespective of vehicle or compound A treatment.

When co-administered with rhBDNF, compound A appeared to yield therapeutic properties against the stroke-induced loss in thalamocortical connectivity. Specifically, a one-way ANOVA reported an overall treatment effect on the density of CTB+ve cells in both the RE (*F*_3,32_ = 7.883, *p* = 0.0004; Figure 4A) and the MD thalamic nuclei (*F*_3,32_ = 4.349, *p* = 0.0112; Figure 4C) of stroke animals. Post-hoc comparisons failed to identify a difference in CTB+ve cell densities within the RE and MD of vehicle-treated stroke animals and stroke animals treated with either compound A or rhBDNF alone. In contrast, stroke animals that received the combined treatment of compound A + rhBDNF (RE: 1274 ± 159; MD: 1096 ± 76) displayed significantly greater CTB+ve cell densities in both thalamic regions, compared to vehicle-treated stroke animals (RE: *p* = 0.0184; MD: *p* = 0.0069). Moreover, animals that received the combination of compound A + rhBDNF also displayed significantly greater densities of CTB+ve cells in the RE relative to stroke animals treated with compound A (*p* = 0.0002), or rhBDNF alone (*p* = 0.0442). Finally, no stroke treatment or interaction effects were observed on the RE and MD thalamic nuclei volumes of these animals (Figure 4B,D, respectively).

### 2.5. Effect of Compound G Treatments on Thalamocortical Connectivity

Consistent with what was seen above, a two-way ANOVA assessing vehicle- and compound G-treated animals reported an overall effect of stroke, but not glycomimetic treatment, on the density of CTB+ve cells in the RE (*F*_1,32_ = 67.09, *p* < 0.0001; Figure 5A) and MD thalamic nuclei (*F*_1,32_ = 58.49, *p* < 0.0001; Figure 5C). Further, post-hoc comparisons confirmed significant stroke-induced reductions in CTB+ve cell density within the RE and MD of vehicle- (RE: *p* < 0.0001; MD: *p* < 0.0001) and compound G-treated animals (RE: sham, 1468 ± 184, stroke, 1127 ± 160, *p* < 0.0001; MD: sham, 1359 ± 98, stroke, 1091 ± 123, *p* = 0.0002). Interestingly, a significant interaction between stroke and compound G treatment was reported on the CTB+ve cell density within the RE (*F*_1,32_ = 4.874, *p* = 0.0350), however post-hoc comparisons failed to confirm any such effects. Conversely, a significant overall effect of compound G treatment was identified on the CTB+ve cell density in the MD thalamus (*F*_1,32_ = 16.34, *p* < 0.0001), with post-hoc comparisons further confirming that a significantly greater MD CTB+ve cell density in compound G-treated stroke animals relative to vehicle-treated control stroke animals (*p* = 0.0002).

When assessing the combinational effect of compound G + rhBDNF a one-way ANOVA identified an overall effect of treatment on the CTB+ve cell density in both the RE (*F*_3,32_ = 5.798 *p* = 0.0031; Figure 5A) and MD thalamic nuclei of stroke animals (*F*_1,32_ = 6.230, *p* = 0.0021; Figure 5C). Post-hoc comparisons further confirmed significantly greater CTB+ve cell densities in the RE of compound G + rhBDNF-treated animals (1282 ± 118), relative to vehicle-treated stroke controls (*p* = 0.0014). No other significant differences were observed in the RE of all stroke animals. A similar effect on CTB+ve cell densities was observed in the MD thalamus of compound G + rhBDNF-treated stroke animals (1062 ± 90), who displayed significantly greater CTB+ve cell densities the MD thalamus relative to vehicle-treated stroke controls (*p* = 0.0156). However, compound G treatment alone appeared to significantly dampen stroke-induced reductions in MD CTB+ve cell densities to a similar extent as compound G + rhBDNF treatment (compound G vs. vehicle, *p* = 0.0038). Finally, no stroke treatment or interaction effects were observed on the RE and MD thalamic nuclei volumes of these animals (Figure 5B,D, respectively).

### 2.6. Effect of Compound A Treatments on Stroke-Induced Reactive Astrogliosis

Previous work has demonstrated a stroke-induced increase in reactive astrogliosis in the frontal cortex that correlates with the degree of spatial working memory in the trial-unique nonmatching-to-location (TUNL) task [8]. Moreover, significant reactive astrogliosis has also been reported in post-mortem tissues collected from stroke patients who developed delayed cognitive impairments [12]. Therefore, we sought to investigate the effect of glycomimetic treatment on stroke-induced reactive astrogliosis by assessing the mean integrated density values (IDV) of GFAP+ve staining in tissue collected 35-days post-stroke (Figure 6 and Figure 7). We first assessed GFAP+ve staining in peri-infarct (PI) tissue immediately adjacent to the infarct in the PFC (i.e., the glial scar). An initial two-way ANOVA assessing GFAP+ve staining in the PI of vehicle- and compound A-treated animals reported a significant overall effect of stroke (*F*_1,33_ = 89.35, *p* < 0.0001), but not compound A treatment or an interaction between these factors (Figure 6A). Post-hoc comparisons further confirmed a significant stroke-induced upregulation in GFAP+ve expression within the PI region of vehicle- (497,644 ± 97,301; *p* < 0.0001) and compound A-treated stroke animals (465,055 ±196,949; *p* < 0.0001), relative to saline-treated shams (99,728 ± 46,939).

To investigate the effect of glycomimetic treatment on reactive astrogliosis in the white matter adjacent to the stroke site, GFAP+ve staining was quantified in regions of the corpus callosum (CC) near the PI zone (Figure 6B). A two-way ANOVA assessing GFAP+ve staining in the CC of vehicle- and compound A-treated animals revealed significant overall effects of stroke (*F*_1,33_ = 19.90, *p* < 0.0001) and compound A treatment (*F*_1,33_ = 9.04, *p* = 0.0015), as well as a significant interaction between these factors (*F*_1,33_ = 6.317, *p* = 0.0170). Post-hoc comparisons further confirmed a significant stroke-induced upregulation in GFAP+ve expression in the CC of vehicle-treated animals (sham: 166,075 ± 38,826; stroke: 339,166 ± 94,798; *p* = 0.0002). In contrast, compound A-treated stroke animals (168,510 ± 104,173) presented GFAP+ve staining in the CC similar to that of saline- and compound G-treated sham animals, but significantly lower than vehicle-treated stroke animals (*p* = 0.0001).

GFAP+ve staining in the infralimbic (IL) region was quantified to assess reactive astrogliosis in grey matter just anterior to the infarct site (Figure 6C). A two-way ANOVA assessing GFAP+ve staining in the IL region of vehicle- and compound A-treated animals revealed a significant overall effect of stroke (*F*_1,33_ = 24.31, *p* < 0.0001), but not compound A treatment or an interaction between these factors. When adjusted for post-hoc comparisons, a significant stroke-induced upregulation in GFAP+ve expression was reported in the IL region of vehicle- (238,363 ± 79,138; *p* = 0.0124) and compound A-treated stroke animals (252,392 ± 117,482; *p* = 0.0042), relative to vehicle-treated shams (114,896 ± 54,225).

One-way ANOVA assessing the combination of compound A + rhBDNF identified a significant overall effect of treatment on GFAP+ve staining in the CC (*F*_1,33_ = 10.99, *p* < 0.0001; Figure 6B), but not PI (Figure 6A) or IL regions (Figure 6C). Post-hoc comparisons confirmed that compound A treatment, with (147,738 ± 71,161) or without rhBDNF, significantly reduced GFAP+ve expression in the CC when compared to animals treated with a vehicle (compound A: *p* = 0.0001; compound A + rhBDNF: *p* = 0.0004) or rhBDNF alone (275,432 ± 42,814; compound A: *p* = 0.0384; compound A + rhBDNF: *p* = 0.0124). Lastly, stroke animals treated with the combination of compound G + rhBDNF displayed significantly less GFAP+ve staining in the CC region compared to treatment with rhBDNF (*p* = 0.0044), but not compound G, alone.

### 2.7. Effect of Compound G Treatments on Stroke-Induced Reactive Astrogliosis

A two-way ANOVA assessing GFAP+ve staining in the PI of vehicle- and compound G-treated animals reported a significant overall effect of stroke (*F*_1,33_ = 143.3, *p* < 0.0001), compound G treatment (*F*_1,33_ = 4.404, *p* = 0.0436) and an interaction between these two factors (*F*_1,33_ = 10.08, *p* = 0.0032; Figure 7A). Post-hoc comparisons further confirmed a significant stroke-induced upregulation in GFAP+ve expression within the PI region of vehicle- (*p* < 0.0001) and compound G-treated stroke animals (359,100 ± 107,927; *p* < 0.0001), relative to saline-treated shams. In addition, relative to treatment with a vehicle, compound G treatment appeared to significantly dampen GFAP+ve expression in the PI of stroke animals *(p* = 0.0034).

A two-way ANOVA assessing GFAP+ve staining in the CC of vehicle- and compound G-treated animals also revealed significant stroke (*F*_1,33_ = 37.98, *p* < 0.0001) compound G treatment (*F*_1,33_ = 27.43, *p* < 0.0001) and interaction effects (*F*_1,33_ = 4.781, *p* = 0.0414; Figure 7B). Post-hoc comparisons confirmed a significant stroke-induced upregulation in GFAP+ve expression within the CC of stroke animals treated with a vehicle (*p* < 0.0001), but not compound G. Moreover, whilst all sham animals displayed similar GFAP+ve expression in the CC, compound G-treated stroke animals (168,510 ± 104,174) displayed significantly less expression in the CC relative to vehicle-treated stroke controls (*p* < 0.0001). However, compound G-treated stroke animals still displayed significantly greater GFAP+ve staining in this region relative to compound G-treated shams (95,783 ± 37,640; *p* = 0.0237).

In line with what was observed in compound A-treated animals, a two-way ANOVA assessing GFAP+ve staining in the IL region of vehicle- and compound G-treated animals revealed a significant overall effect of stroke (*F*_1,33_ = 47.08, *p* < 0.0001) but not compound G treatment or an interaction between these factors (Figure 7C). Supporting this, post-hoc comparisons confirmed a significant stroke-induced upregulation in GFAP+ve expression within the IL region of vehicle- (*p* = 0.0008) and compound G-treated animals (233,180 ± 68,782; *p* < 0.0001), relative to saline-treated shams.

Similar to what was observed in animals treated with compound A + rhBDNF, compound G + rhBDNF treatment provided a similar reduction in GFAP+ve staining to what was observed following compound G treatment alone (Figure 7). A one-way ANOVA identified a significant overall effect of treatment assessing on GFAP+ve expression within the PI of all stroke animals (*F*_3,33_ = 8.885, *p* = 0.0002; Figure 7A). Furthermore, post-hoc comparisons confirmed that compound G, with (298,416 ± 73,236; *p* = 0.0085) or without rhBDNF (*p* = 0.0002), significantly dampened stroke-induced elevations in GFAP+ve expression in the PI, relative to vehicle-treated controls. However, no differences were reported in GFAP+ve staining within the PI region of stroke animals treated with compound G alone or in combination with rhBDNF. A one-way ANOVA assessing GFAP+ve expression within the CC of all stroke animals reported an overall effect of treatment (*F*_3,33_ = 9.312, *p* = 0.0001; Figure 7B). Consistent with what was seen in PI GFAP+ve staining above, post-hoc comparisons confirmed that compound G treatment, with (208,985 ± 58,936; *p* = 0.0019) or without rhBDNF (*p* = 0.0002), significantly dampened the stroke-induced upregulation in GFAP+ve expression in the CC, relative to vehicle-treated control animals. Again, GFAP+ve staining in the CC of stroke animals treated with compound G and compound G + rhBDNF was similar. Lastly, a one-way ANOVA assessing GFAP+ve expression within the IL region of all stroke animals identified an overall effect of treatment (*F*_3,33_ = 4.321, *p* = 0.0112; Figure 7C). However, post-hoc comparisons analyses only reported significantly reduced GFAP+ve expression in the IL region of animals treated with combined compound G + rhBDNF (171,345 ± 35,220) when compared to stroke animals treated with rhBDNF alone (270,829 ± 44,829; *p* = 0.0067).

## 3. Discussion

Over the past decade, advancing technology has allowed chemists to overcome the extremely complex processes and high costs involved with synthesis of glycomimetics [22,23]. Given GAGs play a major role in processes of neuroregeneration and plasticity [14,33,34], the application of glycomimetics has become of significant interest to preclinical research for targeting neurodegenerative diseases. Presently, we report the therapeutic potential of two novel glycomimetics, compounds A and G, demonstrating their ability to modulate thalamocortical connectivity and/or reactive astrogliosis following photothrombotic PFC stroke. Interestingly, despite these pathological mechanisms being impediments to functional recovery after stroke as well as intact spatial memory in healthy rodents [9,10,11,35,36], neither glycomimetic alone were sufficient to facilitate recovery of spatial memory in our stroke model. However, treating stroke animals with the combination of compound G in conjunction with intracerebral rhBDNF successfully improved spatial memory at four weeks post-stroke in the OLRT. In addition, compound G + rhBDNF treatment was the only treatment combination that was able to prevent the stroke-induced loss of thalamocortical connectivity and GFAP+ve reactive astrogliosis across all brain regions assessed in this study.

Preclinical research has routinely demonstrated the role of BDNF treatment in mechanisms of neuroregeneration and functional recovery [37,38,39,40,41]. Furthermore, it is well established that a critical level of BDNF signalling is required for functional recovery to occur in the weeks following experimental stroke [30,31,32,42]. Therefore, it was surprising that we failed to observe any significant effect of rhBDNF treatment on OLRT performance. However, whilst the therapeutic efficacy of exogenous BDNF treatment has been well documented following ischaemic damage to the motor cortex [42,43], to the best of our knowledge, the role of BDNF in promoting cognitive recovery following strokes targeting the PFC has yet to be assessed. Indeed, amounting work supports extensive heterogeneity in the cellular, molecular and transcriptional profiles of different cortical regions of the mammalian brain [44,45,46]. Recent work from Saba and colleagues has further demonstrated neuroprotective properties of exogenous BDNF against 3-nitroprionic acid (3-NP)-induced excitotoxic damage in cultures derived from striatal, but not cortical, astrocytes [47]. It should also be noted that the degree of ischaemic damage acquired following experimental stroke in rodents appears to vary across cortical and subcortical regions [48,49,50,51]. Together, the above literature supports the existence of regional differences in neurobiological function that may have contributed to the lack of therapeutic effects seen following rhBDNF treatment in the present study.

Prior studies have noted the ability of sulphated GAGs to house GFs (including BDNF) within their large negative binding pockets, acting to enhance protein stability and bioavailability, and also facilitate GF-receptor binding [29,52,53]. Our findings support the importance of this interaction, with rhBDNF and compound G being required in combination to overcome damage caused by stroke and to facilitate functional recovery in our PFC stroke model. These findings are consistent with those from Wen and colleagues who assessed the effect of administering a hydrogel embedded with antibodies targeting the Nogo receptor (antiNgR) [54], a receptor that plays a fundamental role in inhibition of axonal growth [55], in a rat model of SCI. Although antiNgR treatment was sufficient to significantly reduce inflammation, astrogliosis and microgliosis within the glial scar, animals displayed the same degree of SCI-induced locomotor impairment to untreated animals. However, when antiNgR-embedded hydrogels were loaded with BDNF- and VEGF-loaded microspheres, animals displayed a similar reduction in astrogliosis and microgliosis, but enhanced angiogenesis and significant recovery of locomotor function. Together, these findings support that our novel glycomimetics acted synergistically with rhBDNF and that a combinational approach is likely required to maximise recovery after stroke.

Early work investigating mechanisms of CNS development has identified the ability of GAGs to regulate axonal guidance and growth [56]. GAGs have since been implicated in mediating a range of regenerative processes throughout the adult brain, including the ability to influence synaptic plasticity through their interactions with GFs and/or via specialised forms of the ECM called perineuronal nets (PNNs) [57,58]. Under pathological conditions such as stroke, increased accumulation of PNNs around GABAergic interneurons has been reported to exacerbate injury-induced behavioural impairments [59]. Further, interventions that degrade PNNs and/or enhance cleavage of their GAG chains have been reported to create conditions favourable for synaptic remodelling and functional recovery [21,60,61,62,63]. PNNs have also been implicated in both the formation and retention of learning and memory processes in rodents [64,65,66,67]. However, the exact role that PNN dysregulation plays in the development of cognitive impairments following stroke to the PFC has not been elucidated, Therefore, future work is required to assess PNN function and modulation thereof using GAGs in the establishment of cognitive impairments.

Reactive astrogliosis is a hallmark feature of secondary degeneration following cortical stroke (including PFC lesions), and is known to contribute to the persistent, delayed degeneration of neural regions distant from the initial site of injury, as well as hinderance of synaptic reorganisation and functional recovery after stroke [35,36,68,69,70]. Consistent with what has been reported previously, we report persistent GFAP+ve reactive astrogliosis both within and surrounding the glial scar, during the chronic phase of stroke [8,71,72]. Pharmacological interventions that dampen stroke-induced upregulation of GFAP+ve protoplasmic astrocytes in the PI regions of the glial scar facilitate functional recovery in rodents [73,74,75]. For instance, the administration of glypican, one-week following MCAo significantly reduces expression of PI GFAP+ve reactive astrocytes and improves sensorimotor function at two-weeks post-stroke [20]. Consistent with these reports, stroke animals treated with the combination of compound G + rhBDNF resulted in a significant decrease in stroke-induced GFAP+ve staining within the PI region, as well as both the adjacent white (CC) and grey matter (IL) regions. However, it is interesting to note that we still demonstrated the ability of compound A or G treatment alone (i.e., without rhBDNF) to affect a significant decrease in GFAP+ve staining within the CC or CC and PI regions, respectively. The fact that these changes were not sufficient to preserve spatial memory performance during the ORLT task in stroke animals, suggest that reactive astrogliosis in both the white and grey matter adjacent to the primary site of stroke may be critical for limiting post-stroke recovery in our PFC stroke model. Moreover, alleviating secondary degeneration in both of these regions is required to promote functional recovery of cognition after stroke. This theory is supporting by Nam and colleagues who demonstrated that reactive astrocytes impose excessive GABA-mediated inhibition on nearby neuronal processes following subcortical white matter stroke [70]. Importantly, the authors further demonstrated that the widespread, pharmacological blockade of astrocytic GABA synthesis using the MAO-B inhibitor, KDS2010, significantly reduced the extent of secondary degeneration in the sensorimotor cortex and facilitated rehabilitation-assisted functional recovery. However, clinical studies contest this theory, instead supporting the association between reactive astrogliosis in the frontal white matter (i.e., CC) and delayed-onset cognitive impairments in humans [12].

It is also interesting that GFAP+ve reactive astrogliosis in the white matter appeared to be sensitive to compound A alone, whereas the grey matter astrocytes were not affected by this treatment. Differences in the transcriptional and morphological phenotypes of grey and white matter astrocytes are thought to underlie differing susceptibilities of astrocyte populations to ischaemic damage in rodents [76,77]. Therefore, due to the heterogeneity of astrocytes within the brain (even in the same region such as cortex) it is conceivable that astrocytes may respond differently to stroke and therapeutic agents (i.e., compound A). In this regard, future work is needed to properly elucidate the molecular composition and role of reactive astrocytes from different brain regions in our PFC stroke model. Furthermore, treatments targeting white matter astrocytes may prove beneficial to reducing astrocyte-mediated retraction of thalamocortical connections, thereby restricting cognitive impairment in our PFC stroke model.

One key difference in the molecular composition of our glycomimetic compounds is the different sulphated patterns of each compound. Previous work has identified that distinct organs or tissues produce unique sulphation and glycosylation patterns on heparan sulphate GAGs [78,79]. This finding, together with the fact that GAGs demonstrate extensive heterogeneity through post-translational modifications, has led to the generation of a complex hypothesis known as the ‘sugar or sulphation code’ [80]. The sugar code hypothesis suggests that particular HS motifs/modifications orchestrate specific cell-ligand interactions and cell signalling pathways in the mammalian CNS. Indeed, mounting evidence has demonstrated the involvement of specific GAG sulphation patterns in mediating GAG-GF and initiation of cell signalling pathways involved in axonal growth, neurogenesis and synaptogenesis [81,82,83,84]. It is therefore possible that the different sulphation patterns between compounds A and G may underlie the differing interactions of these compounds with rhBDNF and also their influence on post-stroke inflammation and thalamocortical connectivity. Further support for this notion has come from recent findings from Hippensteel and colleagues who demonstrated that that the binding affinity of HS fragments to BDNF preferentially increased with sulphation at the 2-*O*-position of IdoA and *N*-position of glucosamine [85].

GAGs are characterised by a staggering molecular heterogeneity, resulting from differences in the length of polysaccharide chains, disaccharide composition and/or post-translational modifications to the glycan chains [86,87,88]. Although accumulating evidence is identifying distinct biological actions of specific GAG variants in the brain [89,90,91,92], our limited knowledge of GAG composition and related bioactivity significantly impedes the biomedical applications of glycans in pathological scenarios such as stroke. Consistent with this idea, we demonstrate differing effects of two glycomimetics on cognitive behaviour, thalamocortical connectivity and reactive astrogliosis following PFC stroke. We hypothesise that this may be an effect mediated in part by the differing sulphation patterns of each glycomimetic; however, there is sparse literature assessing such effects in vivo within naive animals, let alone in animals exposed to experimental stroke. An additional limitation of in vivo glycomimetic research is that compounds such as the glycomimetics investigated presently are likely to undergo chemical modifications by endogenous enzymes (such as glycosidases and sulphatases) present in the vasculature and cells of the CNS [93,94,95]. These modifications are likely to have altered the function of our glycomimetic compounds within the brain, making it difficult to determine the drugs’ exact mechanism of action.

The current study sought to investigate the therapeutic potential of two novel glycomimetic compounds, with or without the co-administration of rhBDNF, in alleviating post-stroke cognitive impairment following PFC stroke. The combination of compound G + rhBDNF was observed to alleviate impairments in post-stroke spatial memory as well as significantly dampening stroke-induced reactive astrogliosis and loss of thalamocortical connectivity. Importantly, compound G or rhBDNF treatment alone was insufficient to alter post-stroke cognitive impairments. These findings provide support for the growing notion that pharmacologically supplementing the brain with a single therapeutic agent may not be enough to overcome the burden of stroke on cognitive function. Rather, synergistically targeting multiple systems and mechanisms for regeneration and repair is likely required to maximise recovery after stroke. These findings help highlight the importance of better understanding GAG and glycomimetic structure and how different modifications (i.e., sulphation) can modulate their effects in the healthy and ischaemic brain. Future work should aim to better elucidate the molecular mechanism associated with our novel glycomimetics, as well as confirming their therapeutic potential on highly-translatable touchscreen-based cognitive tests.

## 4. Methods

### 4.1. Synthesis of Dendritic Cores

“Short-armed” and “long-armed” dendritic cores of glycomimetics were prepared from cheap commercially available starting materials, pentaerythritol and acrylonitrile via Michael reaction. Treatment with concentrated hydrochloric acid followed by a reaction with NHS (*N*-hydroxysuccinimide) and EDC (1-ethyl-3-(3-dimethylaminopropyl)carbodiimide) afforded the preactivated “short-armed” tetramer core. Arm elongation was achieved by coupling with benzyl ester of 8-aminooctanoic acid. The “long-armed” core was subjected to hydrogenolysis and treatment with NHS/EDC to furnish the preactivated “long-armed” tetramer core.

### 4.2. Synthesis of Sugar Fragments

Peracetylated sugars (glucose and maltose) were used in direct glycosylation with 6-*chloroethanol* in the presence of boron trifluoride etherate to afford chloro-glycosides (mono- and disaccharide, respectively). Displacement of the chloro-group led to glycosides with an azido-linker. Zemplen deacetylation followed by reduction of the azides with Raney Nickel under hydrogen, afforded amino-glycosides suitable for coupling reactions to the dendritic cores [96]. Monosaccharide donor and acceptors for the synthesis of GAG fragments were prepared as previously published by Tyler et al. [26]. Glycosylations (1 + 1) and (2 + 2) afforded di- and tetra-saccharide GAG fragments, respectively. These were partially deprotected and amino-groups on linkers were unmasked (by treatment with Zn in AcOH (*N*-Troc), by hydrogenolysis (*N*-Cbz) and by reaction with piperidine (*N*-Fmoc)) for attachment to the dendritic cores.

### 4.3. Synthesis of the Targeted Glycomimetics (Compounds A and G)

Four equivalents of *gluco*-configured disaccharide sugar fragments with an amino-linker were reacted with the “long-armed” preactivated dendritic core. An over-sulfation reaction led to the generation of OVZ/HS14-16Na in excellent yields after chromatography (herein referred to as compound A; Figure 8).

Coupling of four equivalents of GAG fragments with “long-armed” dendritic cores gave tetramer clusters with fully protected fragments attached. Derivatisation steps of removal of the *O*-acetates and sulfation followed by saponification and hydrogenolysis afforded *O*-sulfated *N*-acetylated tetramers (OVZ/HS15-75, herein referred to as compound G; Figure 8) with two sulfate groups per each disaccharide.

Of note, the therapeutic potential of 12 novel glycomimetics were initially assessed in vitro. These data revealed that two glycomimetics, compound A and compound G, demonstrated strong anti-inflammatory effects and were subsequently chosen for investigation in the current study (data not shown).

### 4.4. Animals and Surgical Procedures

All procedures described in this study were carried out in accordance with the guidelines on the care and use of laboratory animals set out by the University of Otago, Animal Research Committee and the Guide for Care and Use of Laboratory Animals (NIH Publication No. 85-23, 1996). Prior to behavioural testing, mice (2–3 months old, male, C57BL/6J) were housed under a 12-h light/dark cycle with *ad libitum* access to food and water. All animals were randomly assigned to a treatment group with use of an online randomization program (http://www.randomization.com, accessed on 1 October 2019). A summary of all group sizes including exclusions for downstream analyses (e.g., behaviour, CTB tracing, GFAP quantification) can be found in Appendix A. All assessments were carried out by observers blind to the treatment group. Focal cerebral infarction in the PFC of mice were induced by photothrombosis, as previously described [6]. Under isoflurane anaesthesia (2–2.5% in O_2_), mice were placed in a stereotactic apparatus and the skull was exposed through a midline incision, cleared of connective tissue and dried. A cold light source (KL1500 LCD, Zeiss) attached to a 40x objective, giving a 2 mm diameter illumination was positioned 1.2 mm anterior to the bregma, and 0.2 mL of rose bengal solution (Sigma-Aldrich; 10 g/L in normal saline) was administered via intraperitoneal (i.p.) injection. After five minutes, the brain was illuminated through the intact skull for 22 min, creating bilateral lesions to the PFC. Aged-matched sham animals received the same surgery as above, with a 0.2 mL injection of saline (i.p.) replacing that of rose bengal.

On day 30 post-stroke, following completion of the object location recognition task (OLRT), animals were again anaesthetised under isoflurane (2–2.5% in O_2_) and placed into a stereotaxic frame with their skulls exposed. A 5 μL Hamilton syringe was loaded with the retrograde tracer, cholera toxin subunit B (CTB), conjugated to Alexa Fluorophores 488 (CTB-488, ThermoFisher, C34775) and positioned to target the left prelimbic (PL) area (2.1 mm AP, 0.25 mm ML, 1.25 mm DV) through a small burr hole created through the skull. A single 0.2 μL injection of CTB-647 was administered at 0.125 μL/min. Following CTB infusion, the needle was left in place for a further two minutes, before being slowly retracted. The skin was glued back together, and the animals were placed on a heating pad to recover before being returned to their home cage.

### 4.5. In Vivo Drug Dosing

The therapeutic potential of two novel glycomimetic compounds, with or without intracerebral hydrogel injections of recombinant human BDNF (rhBDNF), were investigated by administering a single dose via subcutaneously implanted osmotic minipumps (ALZET-1002, DURECT Corporation: Cupertino, CA, USA). Five days post-stroke, animals were anesthetised and placed onto a stereotaxic frame. Under sterile conditions, glycomimetic compounds were dissolved in 0.9% saline (90 μg/mL) and 100 μL of the drug solution was slowly loaded into the loading port of the minipumps via sterile syringes (released at 0.25 μL/hour over 14-day). Loaded-minipumps were inserted (with the loading port facing away from the incision), via a small (<1 cm) incision made between the shoulder blades of the animals, into a subcutaneous pocket made using blunt forceps under the skin.

A hyaluronan/heparan sulfate proteoglycan biopolymer hydrogel (HyStem-C, BioTime Inc., Alameda, CA, USA) was employed to locally deliver rhBDNF or human IgG-Fc (antibody and vehicle control) to the peri-infarct cortex, as described previously [7,42]. A total of 7.5 μL of HyStem-C was impregnated with either human IgG-Fc (5 μg/mL) or rhBDNF (167 μg/mL). HyStem-C was prepared according to the manufacturer’s instructions. In brief, rhBDNF or human IgG-Fc was added to the HyStem-C/Gelin-S mix (component 1 of hydrogel), followed by the addition of Extralink (component 2 of the hydrogel) in a 4:1 ratio. The impregnated HyStem-C mix was injected immediately after preparation into the stroke cavity using a 30-gauge needle attached to a Hamilton syringe at stereotaxic coordinates 1.2 mm AP, 0 mm ML and 0.75 mm DV. During all surgical procedures, mice received Temgesic^®^ (Buprenorphine hydrochloride) as pain relief on the day of surgery as well as the following day. In addition, for minipump implantation procedures topical lidocaine was also given.

### 4.6. Behavioural Assessment

The object location recognition task (OLRT) is widely used to evaluate spatial working memory in rodents [97,98] and has been previously shown to reliably identify delayed-onset impairments in mice exposed to bilateral PFC strokes [6,9]. OLRT testing was conducted at the same time of day at one- and four-week timepoints to minimise variability. All testing was recorded via overhead cameras before being analysed by a blinded researcher on the software TopScan (CleverSys Inc., Reston, VA, USA). On the day prior to OLRT testing, animals were placed into the centre of the OLRT arena (400 × 400 × 200 mm, plexiglass) without any objects and allowed to roam freely for ten minutes to habituate to the arena and testing room. Of note, all animals showed normal gross motor and a lack of anxiety-like behaviour (i.e., time spent in centre of the arena vs. the outside) during this time, consistent with what we have previously published (data not shown) [6,7].

The following day (day 8 and 29 post-stroke) animals underwent the OLRT to evaluate spatial memory [6]. Mice were initially placed in the centre of the arena that contained two identical objects placed in two, neighbouring corners (80 mm from the corner walls) for a period of ten minutes. Immediately following this pre-test phase, mice were returned to their home cage for one-hour and the arenas were cleaned. Mice were then placed back into the arena for a testing period of three-minutes, in which the location of one of the objects was moved into the opposing corner. Object exploration was defined when the mice were pointing towards (within a 20 mm perimeter around the object) and sniffing an object. Periods where the mice were either standing or climbing on the objects were excluded from the final analysis. The duration spent in the novel location was assessed as the ratio between total time interacting with one object relative to the total time interacting with both objects. Consistent with what has previously been performed in our lab, ceramic bear salt-shakers and soft drink cans were used as OLRT objects at one- and four-weeks post-stroke, respectively. Of note, a timeline of all behavioural and surgical events is shown diagrammatically in Figure 9. The arenas and all objects were cleaned with 30% ethanol between behavioural runs to prevent the presence of any confounding odours.

### 4.7. Infarct Volume

Five days following CTB injections (day 35 post-stroke) animals were anesthetised and transcardially perfused with 4% paraformaldehyde (PFA). The brains were removed and postfixed for 1 h in 4% PFA before being transferred to 30% sucrose. The brains were cut into 40 μm thick coronal sections on a sliding microtome with a freezing stage, with all sections stored in cryoprotectant at −20 °C. Infarct volume was determined by histological assessment using a previously published cresyl violet staining protocol [6]. Infarct volume was quantified using ImageJ (National Institutes of Health, Bethesda, Maryland, USA) by an observer blinded to the animal’s treatment groups and was based on obtaining measurements from every 6th section through the entire infarct. Dorsal borders of the infarct were first estimated by tracing the remaining cerebral tissue back to the midline (ensuring a symmetrical curvature across left and right hemispheres), before infarct areas (mm^2^) were calculated. Infarct volume was then quantified as follows:Infarct volume (mm^3^) = area (mm^2^) × section thickness × section interval

### 4.8. Cholera Toxin Subunit B (CTB) Immunofluorescence and Cell Counting

Following tissue collection, every 4th section was mounted onto a gelatin-coated glass slide, air dried, passed sequentially through alcohols (50%, 70%, 95% and 100%) and xylene, before being cover slipped using DPX mounting solution. Regions of interest (ROI; nucleus reuniens, RE; and mediodorsal thalamus, MD) were selected based on previous work illustrating a stroke-induced loss of connectivity between the PFC and these regions in our PFC stroke model [8]. ROI’s were identified using Paxinos and Franklin mouse brain atlas [98] and then imaged using a montaging microscope (Olympus BX51) with a 10× objective. For each mouse, five sections (160 μm apart) were analysed per ROI. Using ImageJ software, each photomicrograph was thresholded to clearly identify CTB+ve cells with minimal background noise. Thresholded photomicrographs were converted into binary images, and the watershed function was used to break apart CTB+ve cell clusters prior to cell counting. CTB+ve cells were then counted using the analyse particle function (cell size, 20–200 μm; circularity, 0.4–1.0) to give a total population of CTB+ve cells. Confirmation of positive cells was also confirmed visually with manual cell counts. To ensure the measured ROIs were consistent across animals, ROI volume was calculated by multiplying the ROI area (mm^2^) by the section thickness and the section interval, and compared across all animals. Lastly, CTB+ve cell densities were calculated for each animal by dividing the CTB+ve population totals by the ROI volume.

### 4.9. Glial Fibrillary Acid Protein (GFAP) Immunocytochemistry

Immunofluorescent labelling of GFAP was performed to assess the degree and spread of reactive astrogliosis. Briefly, every sixth section was washed thoroughly in tris-buffered saline (TBS: 3 × 10 min) and incubated for 48 h at 4 °C in primary antibody solution consisting of polyclonal chicken anti-GFAP antibody (dilution 1:2000; Millipore, #AB5541) and TBS containing 0.3% Triton X-100 and 2% normal goat serum (NGS). Sections were then washed three times in TBS (10 min per wash) before incubation in a goat anti-chicken 550 secondary antibody (Thermofisher, #SA5-10071) for two hours at room temperature. After subsequent washing in TBS (3 × 10 min), sections were mounted onto gelatine-coated glass slides, air-dried, passed sequentially through alcohols (50%, 70%, 95%, 100%) and xylene, before being coverslipped using DPX mounting solution. Photomicrographs were taken using an Olympus BX51 microscope set with a 10× objective lens. Using the software FIJI (National Institutes of Health, Bethesda, Maryland, USA), the integrated density value (IDV) was measured across four ROI (100 × 200μm rectangles), as previously published [8]: the peri-infarct cortex (PI, approximately cortical layer 2/3), corpus callosum (CC), infralimbic (IL) and intact unstained PFC as a background control region. Specifically, the background IDV was subtracted from the IDV of the other three ROI, and a mean IDV was calculated for each of these three ROI (PI, CC and IL) by averaging the resulting IDV values (three sections per ROI/animal). Of note, the IL and CC were chosen to represent white and grey matter regions sitting immediately adjacent to the peri-infarct area.

### 4.10. Statistical Analyses

All animals within this study were tested as a single experimental cohort of mice. However, as compound A and G are not known to interact with one another and are structurally different, each glycomimetic compound was assessed independently via a two-step approach. First, the effect of each glycomimetic treatment itself in sham and stroke animals were compared to vehicle-treated controls and evaluated by a two-way ANOVA, followed by post-hoc multiple comparisons. The effects of co-administering glycomimetic treatment with rhBDNF were then assessed by a one-way ANOVA followed by post-hoc comparisons, comparing all stroke animals only. The only exception to this approach was for analysing infarct volumes, for which a one- and two-way ANOVA were employed to assess total infarct volumes and to assess for asymmetrical differences in infarction across all stroke animals, respectively. It should also be noted that descriptive statistical data from vehicle-treated sham and stroke animals have only been reported in text during the first presentation of such data (i.e., compound A sections), not in the repeated sections (i.e., compound G sections). All data are presented as mean ± S.D.

## Figures and Tables

**Figure 1 ijms-23-04817-f001:**
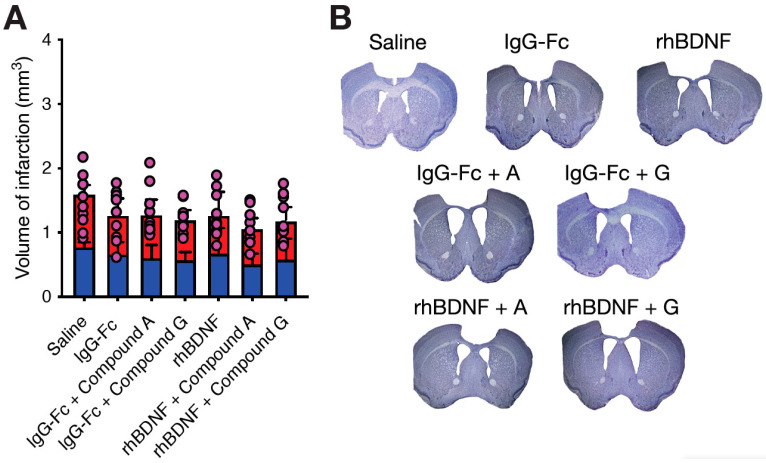
Quantification of infarct volumes at day-35 post-stroke. Assessment of infarct volumes found no significant differences in total infarct volumes (pink dots, (**A**)), regardless of saline (*n* = 10), IgG-Fc (*n* = 10), IgG-Fc + compound A (*n* = 10), IgG-Fc + compound G (*n* = 10), rhBDNF (*n* = 9), compound A + rhBDNF (*n* = 9) or compound G + rhBDNF treatment (*n* = 9). Similarly, no asymmetrical differences were observed in hemispheric damage across all stroke animals (left hemisphere, blue bar; right hemisphere, red bar; (**A**)). Representative PFC strokes are shown for all treatment groups in (**B**), including a section from a saline-treated sham animals for comparison.

**Figure 2 ijms-23-04817-f002:**
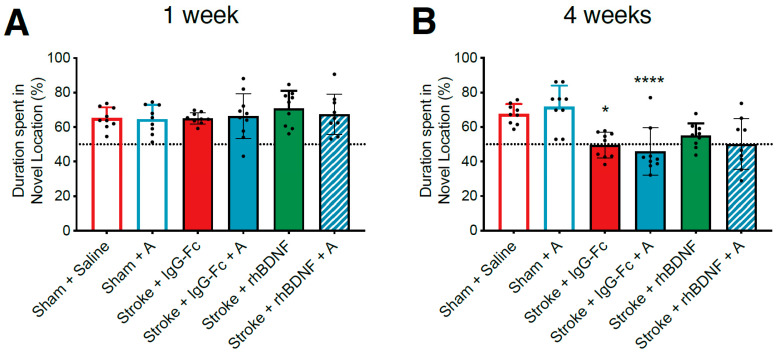
Effects of compound A treatment, with or without rhBDNF, on object location recognition task performance (OLRT) following PFC stroke. OLRT performance at one- (**A**) and four-weeks post-stroke (**B**) was compared between sham and stroke animals treated with vehicle (sham + saline, *n* = 9; stroke + IgG-Fc, *n* = 9), compound A (sham: *n* = 9; stroke: *n* = 10), rhBDNF (stroke only, *n* = 9) or compound A + rhBDNF treatment (stroke only, *n* = 9). At one-week post-stroke, all animals spent a similar percentage of the testing duration exploring the object in the novel location. In contrast, at four-weeks post-stroke, stroke animals treated with compound A or a vehicle solution spent significantly less time exploring the object in the novel location relative to their respective sham counterparts, indicating a delayed-onset impairment in spatial memory. However, one-way ANOVAs comparing the effect of co-administering rhBDNF with compound A failed to find any difference in the performance of all stroke animals at both timepoints. Two-way ANOVA: * *p* < 0.05, **** *p* < 0.0001, relative to saline-treated sham performance.

**Figure 3 ijms-23-04817-f003:**
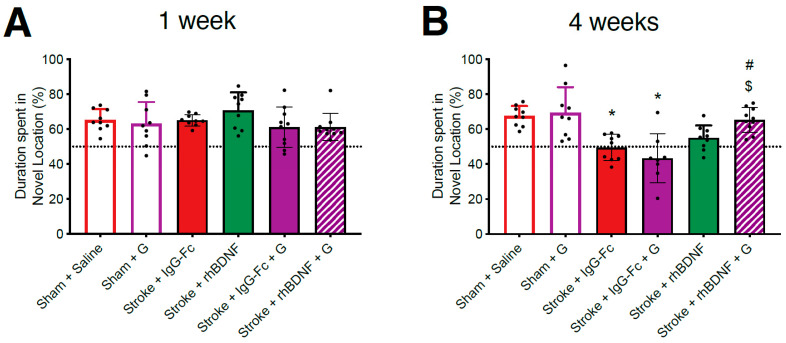
Effects of compound G treatment, with or without rhBDNF, on object location recognition task (OLRT) performance following PFC stroke. OLRT performance at one- (**A**) and four-weeks post-stroke (**B**) was compared between vehicle (sham + saline, *n* = 9; stroke + IgG-Fc, *n* = 9), compound G (sham: *n* = 9; stroke: *n* = 9), rhBDNF (stroke only, *n* = 9) or compound G + rhBDNF (stroke only, *n* = 9). At one-week post-stroke, all animals spent a similar percentage of the testing duration exploring the object in the novel location. However, at four-weeks post-stroke animals treated with compound G or a vehicle solution spent significantly less time exploring the object in the novel location relative to their respective sham counterparts, indicating a delayed-onset impairment in spatial memory. In addition to this, a one-way ANOVA comparing performance of all stroke animals reported an overall effect of treatment, with post-hoc comparisons confirming that stroke animals treated with compound G + rhBDNF spent significantly more time investigating the object in the novel location relative to stroke animals treated with either a vehicle solution- or compound G alone. Two-way ANOVA: * *p* < 0.05, relative to saline-treated sham performance. One-way ANOVA: $ *p* < 0.05, relative to vehicle-treated stroke controls. # *p* < 0.05, relative to compound G-treated stroke animals.

**Figure 4 ijms-23-04817-f004:**
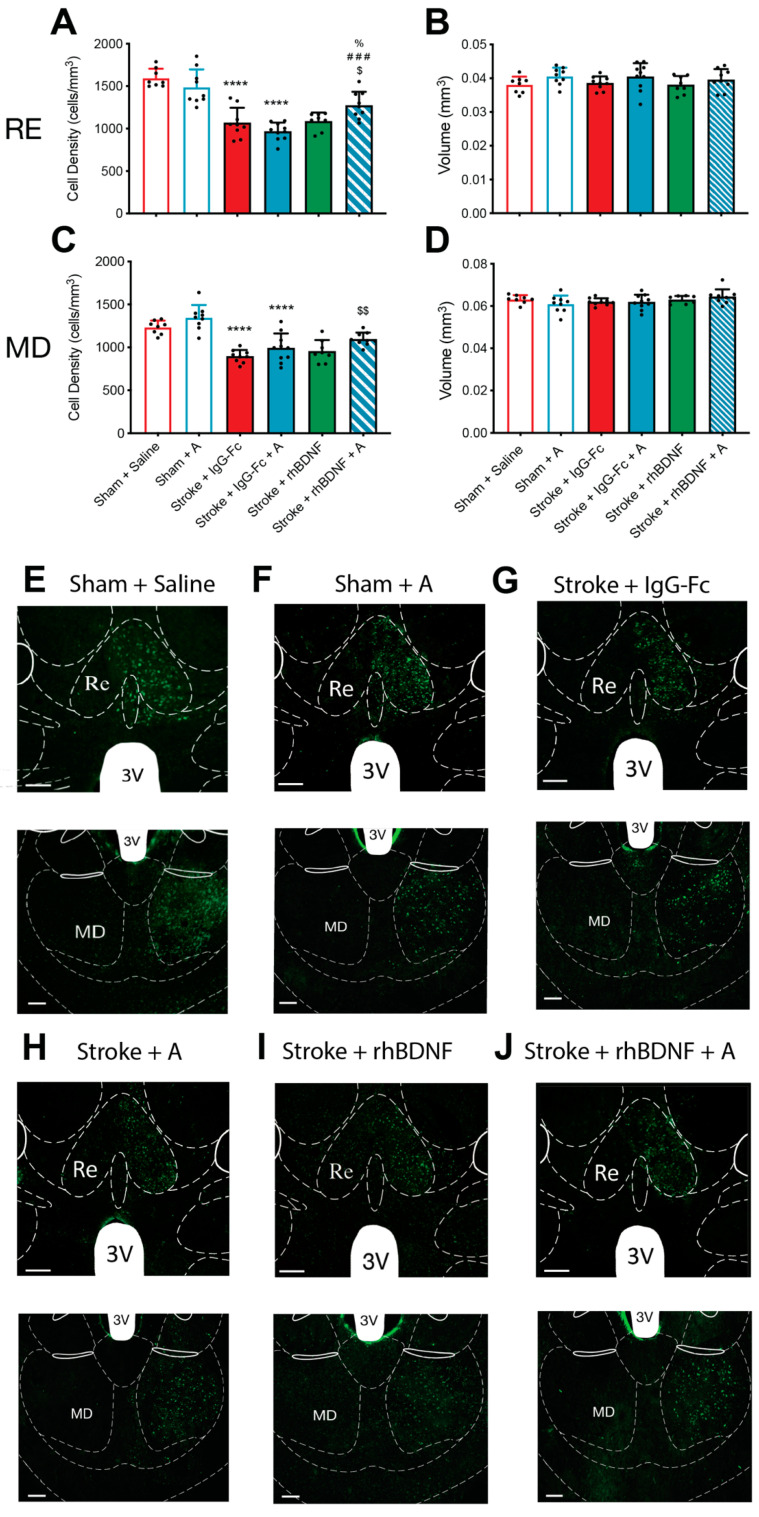
Effects of compound A treatment, with or without rhBDNF, on cholera toxin subunit B (CTB)+ve cell populations in the thalamus following PFC stroke. CTB+ve staining was quantified in the nucleus reuniens (RE, −1.34 mm bregma) and mediodorsal thalamus (MD, −1.34 mm bregma) at day-35 post-stroke. Stroke caused a significant reduction in the density of CTB+ve cells in the RE (**A**) and MD thalamus (**C**) of animals treated with a vehicle (sham + saline, *n* = 8; stroke + IgG-Fc, *n* = 8) or compound A (sham, *n* = 9; stroke, *n* = 10). Compound A failed to affect these CTB+ve cell densities in sham and stroke animals. Furthermore, combinational treatment with compound A + rhBDNF (*n* = 9) significantly dampened the stroke-induced reduction in the density of CTB+ve cells within the RE and MD thalamus compared to all other experimental treatments. Lastly, no difference was observed in the volume of the RE (**B**) or MD thalamus (**D**) of all animals. Representative images of both thalamic regions are shown in (**E**–**J**). Two-way ANOVA: **** *p* < 0.0001, relative to vehicle-treated shams; One-way ANOVA: $ *p* < 0.05, $$ *p* < 0.01, relative to IgG-Fc; ### *p* < 0.001, relative to compound A; % *p* < 0.05. 3 V = third ventricle. Scale bar (white bar, bottom left) = 100 μm.

**Figure 5 ijms-23-04817-f005:**
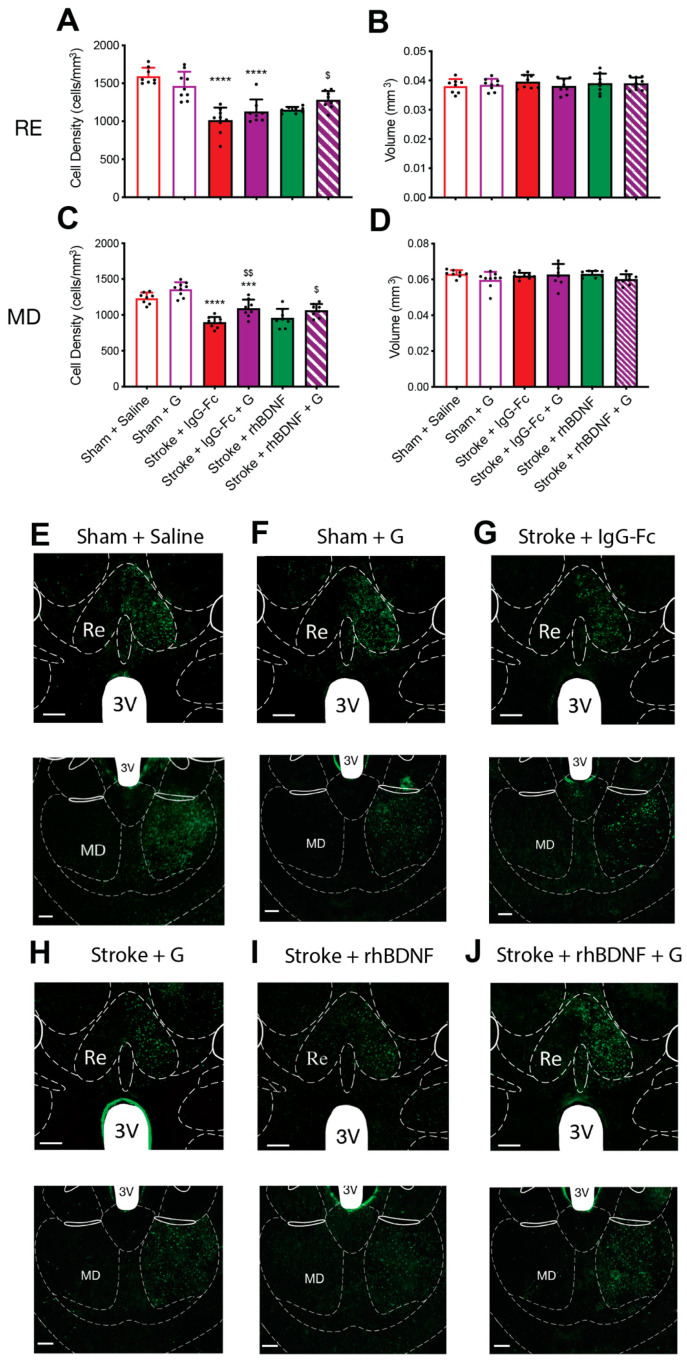
Effects of compound G treatment, with or without rhBDNF, on cholera toxin subunit B (CTB)+ve cell populations in the thalamus following PFC stroke. CTB+ve staining was quantified in the nucleus reuniens (RE, −1.34 mm bregma) and mediodorsal thalamus (MD, −1.34 mm bregma) at day-35 post-stroke. Stroke-caused a significant reduction in the density of CTB+ve cells in the RE (**A**) and MD thalamus (**C**) of animals treated with a vehicle (sham + saline, *n* = 8; stroke + IgG-Fc, *n* = 8) or compound G (sham, *n* = 9; stroke, *n* = 8). Compound G treatment significantly reduced the stroke-induced loss in the density of CTB+ve cells in the MD thalamus only. Further, the combinational treatment of compound G + rhBDNF (*n* = 7) significantly dampened the stroke-induced loss of CTB+ve cell densities in the RE and MD thalamus, but to a similar extent as compound G treatment alone. Lastly, no difference was observed in the volume of the RE (**B**) or MD thalamus (**D**) across all animals. Representative images of both thalamic regions are shown in (**E**–**J**). Two-way ANOVA: *** *p* < 0.001, **** *p* < 0.0001, relative to sham counterpart; $$ *p* < 0.001, relative to vehicle-treated counterpart; one-way ANOVA: $ *p* < 0.05, relative to IgG-Fc. 3 V = third ventricle. Scale bar (white bar, bottom left) = 100 μm.

**Figure 6 ijms-23-04817-f006:**
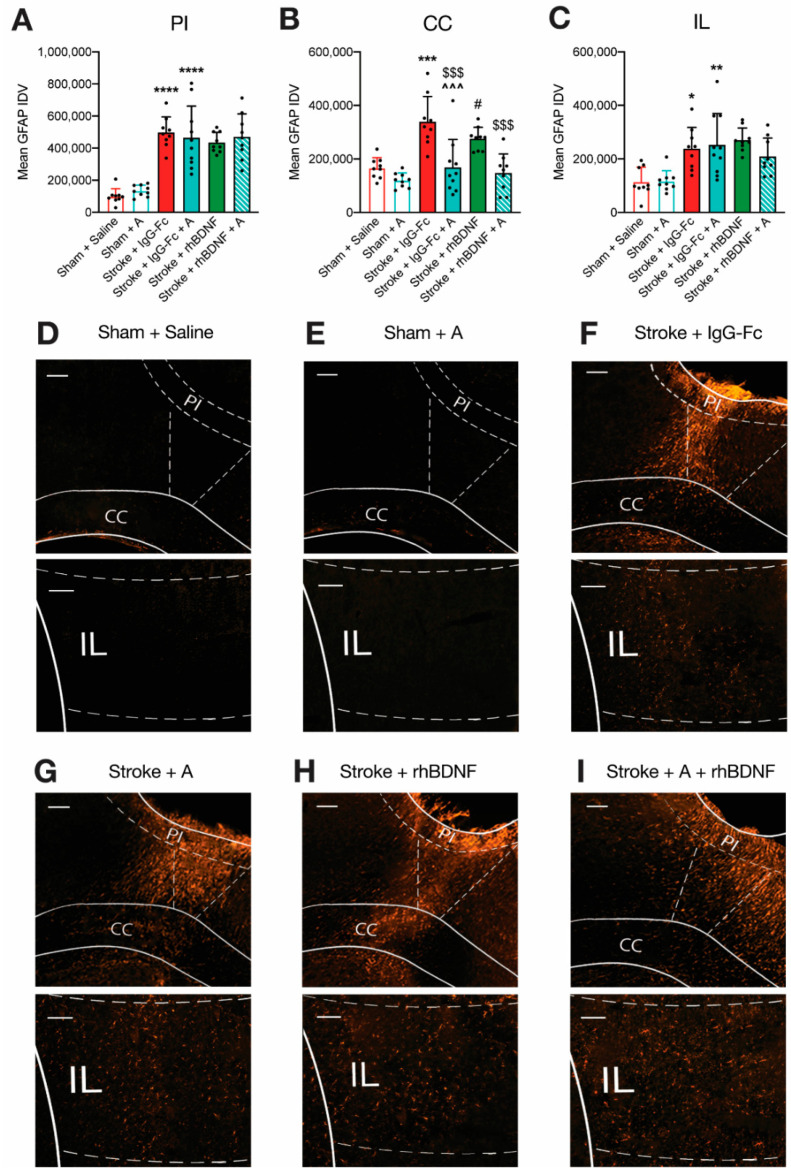
Effects of compound A treatment, with or without rhBDNF, on stroke-induced reactive astrogliosis. Bilateral strokes to the PFC induced a significant upregulation in the mean integrated density value (IDV) of GFAP+ve staining in the peri-infarct (PI, (**A**)), corpus callosum (CC, (**B**)) and infralimbic (IL, (**C**)) regions of vehicle-treated animals (sham + saline, *n* = 9; stroke + IgG-Fc, *n* = 9). A similar stroke-induced upregulation in GFAP+ve expression was observed in the PI and IL regions of compound A-treated animals (sham, *n* = 9; stroke, *n* = 10). In contrast, compound A treatment, with (*n* = 9) and without rhBDNF (*n* = 8), appeared to dampen stroke-induced elevations in GFAP+ve expression within the CC. Representative micrographs of GFAP+ve staining are shown in (**D**–**I**). Two-way ANOVA: * *p* < 0.05, ** *p* < 0.01, *** *p* < 0.001, **** *p* < 0.0001, relative to saline-treated sham animals. ^^^ *p* < 0.001, relative to vehicle-treated stroke animals. # *p* < 0.05, relative to COMPOUND G + rhBDNF-treated stroke animals. $$$ *p* < 0.001, relative to vehicle-treated stroke animals. Scale bar (white bar, top left) = 100 μm.

**Figure 7 ijms-23-04817-f007:**
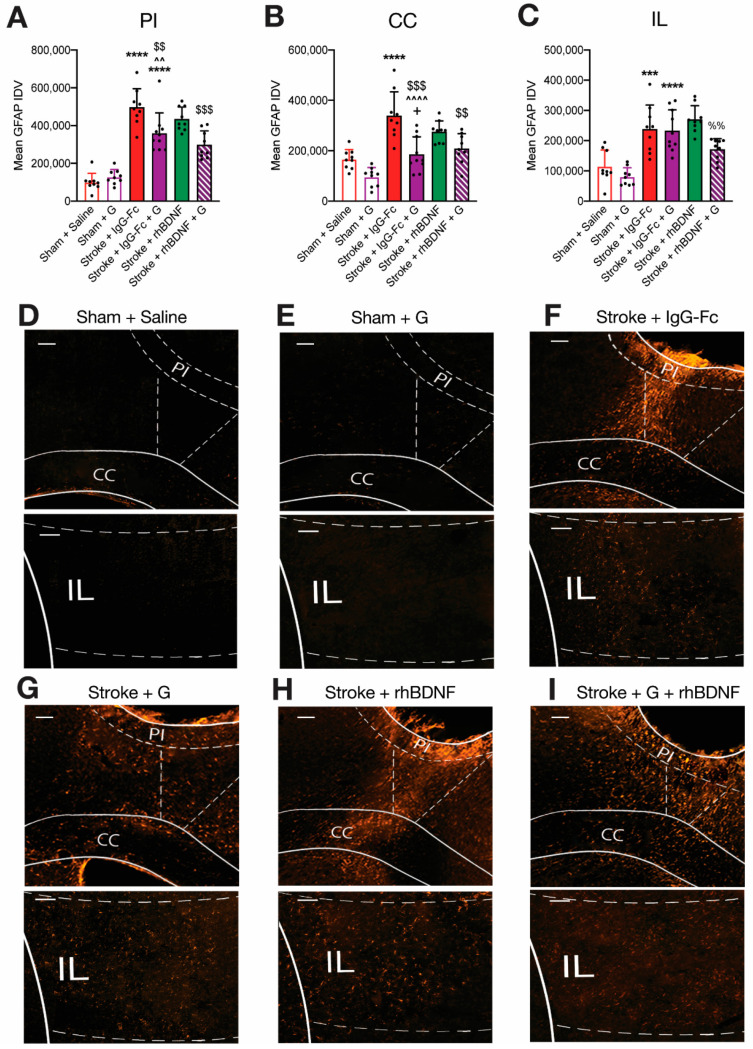
Effects of compound G treatment, with or without rhBDNF, on stroke-induced reactive astrogliosis. PFC strokes induced a significant upregulation in the mean integrated density value (IDV) of GFAP+ve expression in the peri-infarct (PI, (**A**)), corpus callosum (CC, (**B**)) and infralimbic (IL, (**C**)) regions of vehicle-treated animals (sham + saline, *n* = 9; stroke + IgG-Fc, *n* = 9). Compound G treatment alone had no effect in sham animals (*n* = 9) but significantly dampened stroke-induced elevations in GFAP+ve expression in the PI and CC regions of stroke animals (*n* = 9). Moreover, compound G + rhBDNF treatment (*n* = 9) further caused a reduction in GFAP+ve expression within IL regions (**D**–**I**). Two-way ANOVA: *** *p* < 0.001, **** *p* < 0.0001, relative to saline-treated sham animals. ^^ *p* < 0.01, ^^^^ *p* < 0.0001, relative to vehicle-treated stroke animals; one-way ANOVA: $$ *p* < 0.01, $$$ *p* < 0.001, relative to vehicle-treated stroke animals; %% *p* < 0.01, relative to rhBDNF-treated stroke animals. Scale bar (white bar, top left) = 100 μm.

**Figure 8 ijms-23-04817-f008:**
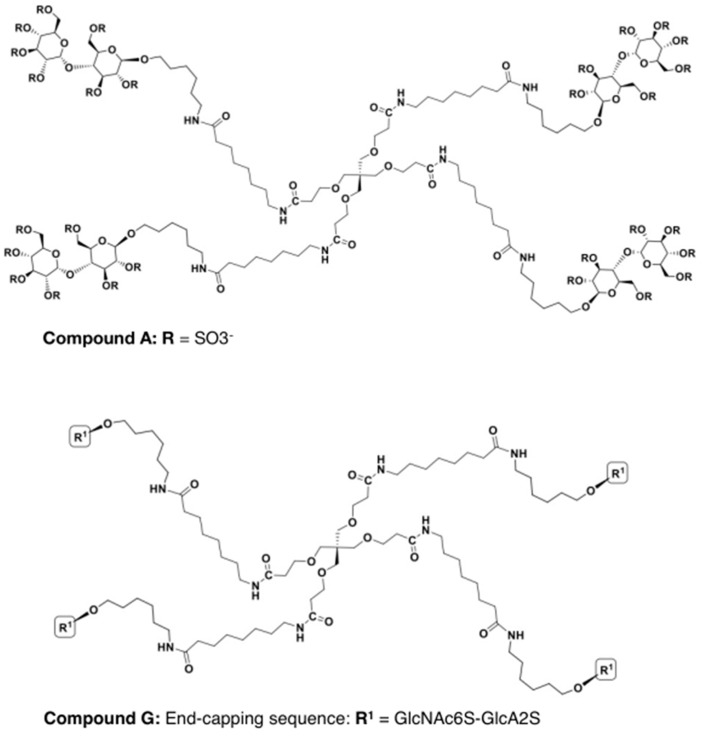
Chemical structure and composition of the novel glycomimetics, compound A and G. Compound A consisted of a tetramer with over-sulfated maltose residues, whereas compound G consisted of an octa-sulfated heparan sulfate tetramer attached to a PET core. A more detailed description of the generation of these compounds can be found in [26].

**Figure 9 ijms-23-04817-f009:**
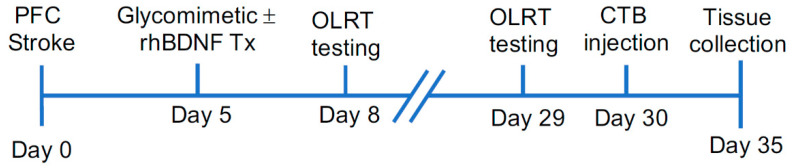
Experimental timeline illustrating the sequence of behavioural and surgical events following photothrombotic PFC stroke. Animals were pseudorandomised into treatment groups prior to receiving either PFC stroke or sham surgeries. After receiving either glycomimetic (compound A or G) or vehicle (saline) osmotic minipumps, with or without intracerebral injections of rhBDNF or vehicle (IgG-Fc), animals underwent the object location recognition task (OLRT) at one- and four-weeks post-stroke. Prior to tissue collections, animals received prelimbic injections of the retrograde cholera toxin subunit B (CTB) to allow for tracing of thalamocortical connections.

## Data Availability

All data is stored on managed institutional servers and will be made available upon request to the corresponding author.

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
