# Peer review of "Recovery of Post-Stroke Spatial Memory and Thalamocortical Connectivity Following Novel Glycomimetic and rhBDNF Treatment"

_ijms, 2022, doi:10.3390/ijms23094817_

Round 1

Reviewer 1 Report

Recovery of post-stroke spatial memory and thalamo-coritical connectivity with glycomimetics

This paper presents evidence in mice for partial recovery by locally administered glycomimetics, of reduced spatial memory and thalamo-cortical connectivity following induced photo-thrombosis in vessels of the pre-frontal cortex.  The paper is well-written, and the analysis seems appropriate.  

Main Points

1). The interpretation of partial recovery of spatial memory relies on observation over a single 3 minute period of the behaviour of each mouse in the OLRT (object location recognition task).  For a reader less-familiar with the field, the authors need to justify the validity of this test as truly reflecting spatial memory (rather than, say, the mouse having more interest in the most fresh scent from the most recently-handled object). At the least, they need to cite (in section 2.6) original sources which refer to validation of this test, and perhaps also in the introduction.  

2). The image quality for Figures 6 and Figures 7 (for the charts and their labelling, and the photographs and their labelling) is poor, being unacceptably out-of-focus, and for the Fig 6 photographs having poor contrast. These must be improved if possible. 

All other comments relate to minor typos/rephrasing etc., requiring correction:   These include:

  1. Line 67-70: The use of ‘ie’ in line 69 could imply that BDNF is the only GF to which GAGs can bind although the text indicates: ‘GAGs….bind to GFs’.  The use of  ‘eg’ rather than ‘ie’ would seem better here. Also, in writing:  ‘GAGs.…act as…co-receptors and bind to GFs….to their respective receptors’ ,  do the authors mean that there are specific GAG receptors on GFs, or should it be: ‘GAGs.…act as…co-receptors and help bind GFs….to their respective receptors’
  2. Line 87: ‘ glycosylation ’  (or possibly ‘ glycosylations ’) but not with the apostrophe.
  3. Line 82,88,91,94: Is ‘afford’ /’afforded’ the correct word here ?. Do the authors mean ‘affix’/’affixed’
  4. Line 640: ‘to afford’ . Should this be:  ‘to effect’
  5. Line 109: ‘compound A and compound’ should have a G added ?
  6. Line 110-111: ‘chosen to investigate’ would be better as :  ‘chosen for investigation’
  7. Line 116: ‘can be found in 25’ This would read better as : ‘can be found in Tyler et al25 ‘  or:  ‘was published by Tyler et al25 ‘ 
  8. Line 122: ‘all aged animals randomly assigned’ . What does this mean ?  Is it :  ‘all adult animals above xx wks of age were randomly assigned ?’  Please clarify and correct.
  9. Line 124-5: ‘Focal stroke to the PFC of ….mice were induced’ This would seem better written as a pathophysiologic process :  ie.  ‘Focal cerebral infarction in the PFC of….mice was induced’
  10. Line 129: ‘Bregma’ should be ‘the bregma’
  11. Line 153-4: ‘Loaded pumps when placed port side facing away’ Do the authors mean:  ‘Pumps were loaded when the infusion port was facing away’   or:  ‘Loaded pumps were placed with the infusion port facing away’.  ‘Port side’ might be interpreted as left-hand side (as for a ship).  Please amend accordingly.
  12. Line 172: ‘The days prior to OLRT testing’ was this for the several days before testing, and if so was it standardised for each animal.  Or should it be:  ‘On the day prior to OLRT testing’  ?  Please amend.
  13. Line 187: ‘spent’ rather than ‘spend’
  14. Line 191-2: ‘a timeline…..events is shown’ , rather than ‘are shown’
  15. Line 258: ‘The effect…….was then assessed’ or ‘The effects……were then assessed’ , rather than ‘The effect….were’.
  16. Line 265 ‘All data is presented’
  17. Line 500: ‘reported’ implies ‘publicised’. This would read better as : ‘A two-way ANOVA….found’  or ‘A two-way ANOVA…..revealed’  
  18. Line 591 ‘cortical and subcortical region of 49-52.’ This sentence needs completion.
  19. Line 640 (see points 5 & 6 above)
  20. Line 679 ‘these compounds differing interactions’ , should be either apostraphied: ‘ compounds’ ‘ or better as: ‘the differing interactions of these compounds’
  21. Line 690: ‘glycans’ (without any apostrophe)
  22. Line 694: ‘however sparse literature has assessed’ would be better as: ‘however there is sparse literature assessing’
  23. Line 700: ‘drugs’ should have an apostrophe. Ie. either ‘ drug’s ’ or ‘ drugs’ ‘
  24. Line 707: ‘These findings support for’ should be:  ‘These findings provide support for’
  25. Line 711: ‘help build highlight’ needs rephrasing.  Perhaps just:  ‘help highlight’

Author Response

Reviewer 1:

This paper presents evidence in mice for partial recovery by locally administered glycomimetics, of reduced spatial memory and thalamo-cortical connectivity following induced photo-thrombosis in vessels of the pre-frontal cortex.  The paper is well-written, and the analysis seems appropriate.  

Main Points

1). The interpretation of partial recovery of spatial memory relies on observation over a single 3 minute period of the behaviour of each mouse in the OLRT (object location recognition task).  For a reader less-familiar with the field, the authors need to justify the validity of this test as truly reflecting spatial memory (rather than, say, the mouse having more interest in the most fresh scent from the most recently-handled object). At the least, they need to cite (in section 2.6) original sources which refer to validation of this test, and perhaps also in the introduction.  

The initial description of the OLRT task within the methods has been amended to start with the following: “The object location recognition task (OLRT) is widely used to evaluate spatial working memory in rodents 34, 35 and has been previously shown to reliably identify delayed-onset impairments in mice exposed to bilateral PFC strokes 5, 8.”

We have also added the following – “The arena’s and all objects were cleaned with 30% ethanol between behavioural runs to prevent the presence of any confounding odours.”

We have also added the following to the final sentence of the main manuscript:              “Future work should aim to better elucidate the molecular mechanism associated with our novel glycomimetics, as well as confirming their therapeutic potential on highly-translatable touchscreen-based cognitive tests.”

2). The image quality for Figures 6 and Figures 7 (for the charts and their labelling, and the photographs and their labelling) is poor, being unacceptably out-of-focus, and for the Fig 6 photographs having poor contrast. These must be improved if possible. 

We apologise for this issue as the figures looked fine on our computers.  We have now increased the size of the photomicrographs and saved the figures in higher dpi to address this issue. 

All other comments relate to minor typos/rephrasing etc., requiring correction:   These include:

  1. Line 67-70: The use of ‘ie’ in line 69 could imply that BDNF is the only GF to which GAGs can bind although the text indicates: ‘GAGs….bind to GFs’.  The use of  ‘eg’ rather than ‘ie’ would seem better here. Also, in writing:  ‘GAGs.…act as…co-receptors and bind to GFs….to their respective receptors’ ,  do the authors mean that there are specific GAG receptors on GFs, or should it be: ‘GAGs.…act as…co-receptors and help bind GFs….to their respective receptors’

Changes– “Indeed, GAGs themselves have been observed to act as medium-affinity co-receptors to help bind GFs (e.g. brain-derived neurotrophic factor, BDNF) to their respective receptors

  1. Line 87: ‘ glycosylation ’  (or possibly ‘ glycosylations ’) but not with the apostrophe.

Changes – “…were used in direct glycosylation with…

  1. Line 82,88,91,94: Is ‘afford’ /’afforded’ the correct word here ?. Do the authors mean ‘affix’/’affixed’

We have double checked with the chemists on what was written and ‘afford’ is correct here.

  1. Line 640: ‘to afford’ . Should this be:  ‘to effect’

Changed “to effect

  1. Line 109: ‘compound A and compound’ should have a G added ?

Changed – “… compound  and compound G…”

  1. Line 110-111: ‘chosen to investigate’ would be better as :  ‘chosen for investigation’

Changed – “… chosen for investigation…”

  1. Line 116: ‘can be found in 25’ This would read better as : ‘can be found in Tyler et al25 ‘  or:  ‘was published by Tyler et al25 ‘ 

Changed.- “…as previously published by Tyler et al25.”

  1. Line 122: ‘all aged animals randomly assigned’ . What does this mean ?  Is it :  ‘all adult animals above xx wks of age were randomly assigned ?’  Please clarify and correct.

General information regarding the mice was specified in the sentences prior to the one of question (“Prior to behavioural testing, mice (2-3 months old, male, C57BL/6J) were housed…”), and randomisation was clarified in the statement, “with use of an online randomization program (http://www.randomization.com)”. The term “Aged” was a typo and has been removed from that sentence.

  1. Line 124-5: ‘Focal stroke to the PFC of ….mice were induced’ This would seem better written as a pathophysiologic process :  ie.  ‘Focal cerebral infarction in the PFC of….mice was induced’

Changed – “Focal cerebral infarction in the PFC of mice was induced..

  1. Line 129: ‘Bregma’ should be ‘the bregma’

Changed to “…the bregma…”

  1. Line 153-4: ‘Loaded pumps when placed port side facing away’ Do the authors mean:  ‘Pumps were loaded when the infusion port was facing away’   or:  ‘Loaded pumps were placed with the infusion port facing away’.  ‘Port side’ might be interpreted as left-hand side (as for a ship).  Please amend accordingly.

Authors have now acknowledged prior to this statement that there is a ‘loading site’ on the minipumps (“…drug solution was slowly loaded into the loading sites of the minipumps…”). Remaining paragraph was also altered as follows, “Loaded-minipumps were inserted (with the loading port facing away from the incision), via a small (<1cm) incision made between the shoulder blades of the animals, into a subcutaneous pocket made using blunt forceps under the skin. Loaded pumps when placed port side facing away from the incision before the incision site was closed.”

  1. Line 172: ‘The days prior to OLRT testing’ was this for the several days before testing, and if so was it standardised for each animal.  Or should it be:  ‘On the day prior to OLRT testing’  ?  Please amend.

Changed.- “On the day prior…”

  1. Line 187: ‘spent’ rather than ‘spend’

Changed to “spent

  1. Line 191-2: ‘a timeline…..events is shown’ , rather than ‘are shown’

Changed to “is

  1. Line 258: ‘The effect…….was then assessed’ or ‘The effects……were then assessed’ , rather than ‘The effect….were’.

Changed to “…The effects of co-administering glycomimetic treatment with rhBDNF were…

  1. Line 265 ‘All data is presented’

Changed to “All data is presented…

  1. Line 500: ‘reported’ implies ‘publicised’. This would read better as : ‘A two-way ANOVA….found’  or ‘A two-way ANOVA…..revealed’  

We disagree with the reviewer on this point. Reported has been used throughout the results section and is commonly used in publications to discuss ANOVA effects.

  1. Line 591 ‘cortical and subcortical region of 49-52.’ This sentence needs completion.

Changed – “…across cortical and subcortical regions of….

  1. Line 640 (see points 5 & 6 above)

Changed “…to effect…”

  1. Line 679 ‘these compounds differing interactions’ , should be either apostraphied: ‘ compounds’ ‘ or better as: ‘the differing interactions of these compounds’

Changed to “…underlie the differing interactions of these compounds with rhBDNF…

  1. Line 690: ‘glycans’ (without any apostrophe)

Changed to “…glycans…”

  1. Line 694: ‘however sparse literature has assessed’ would be better as: ‘however there is sparse literature assessing’

Changed to “…however there is sparse literature assessing…”

  1. Line 700: ‘drugs’ should have an apostrophe. Ie. either ‘ drug’s ’ or ‘ drugs’ ‘

Changed to “…drugs…”

  1. Line 707: ‘These findings support for’ should be:  ‘These findings provide support for’

Changed to “These findings provide support for…”

  1. Line 711: ‘help build highlight’ needs rephrasing.  Perhaps just:  ‘help highlight’

Changed to “…help build highlight…

Reviewer 2 Report

This is an interesting paper reporting beneficial spatial memeory deficits and loss of thalamo cortical connectivity resulting stroke in mice. The experimental design and methodology appear sound, however the statistical analysis is not justified and the quality of the figures is so poor one could not determine if the conclusions are supported by the results.

Other more minor points:

The exclusive use of males is not justified other. Women do get stroke.

The occasional sentence is uninterpretable.

Comments on these issues are detialed above in order of appearance in the MS.

Intro:

“Indeed, GAGs have been  observed to act as medium-affinity co-receptors and bind to GFs ( ..) to their respective receptors”. What does this sentence mean?

Methods

Michael’s reaction and similar needs to be referenced if not described in detail.

P3 line 109 “compound A and compound, ??? demonstrating…Presumably conpound G?

Animals:

Animals need to be described first,  in terms of strain, sex, age, or weight. This is also a good place to mention the total number of animals included in all of the studies described.

Line 122 “All aged animals randomly assigned to a treatment group”. What does this mean?

“A single prolonged dose” is not a good choice of words…

“Loaded pumps when placed port side facing away from the incision before the incision site was closed.”  Please rephrase

“IgG-Fc_(5μg/mL)_or_rhBDNF_(0.167μg_/μL)_” – are the concetration units correct? If yes, why the big difference in concentration between active and control? 

What is temgesic?

FIJI and Image J are the same software (Fiji is Image J…) but the software is referred to as “image J” in one place and” FIJI Image J”  in another . This is confusing.

It is not clear how infarct volume was measured, considering the infarcted area is not enclosed by tissue ( as would be the case with a striatal infarct). Under these circumstances, the dorsal ”borders “ of the infarct can not be seen and the volume of the infarct can not be measured directly. There are different solutions to this problem but it it’s not clear what the authors have done.

The rationale for the described  statistical analysis is not clear.

Results

Fig 3. Is very confusing, stacking the red and blue bars on top of each other to denote right and left hemisphere is very confusing. This figure needs to be redrawn.

It is not clear what is meant by “multiple comparisons”, do the authors mean posthoc comparisons?

The results of the  statistical analysis are not detailed enough – for both 2 way and one way ANOVA the main effects (F, p)need to be reported first, prior to results of posthoc anslaysis.  In the absence of a main effect or interaction, posthoc comparisons are not justified.

“compared to animals treated with vehicle-treated stroke animals”-needs fixing

 Figure 6 is of such bad quality /small size it is uninterpretable. Same for figure 7, 8 and 9.

Author Response

Reviewer 2:

This is an interesting paper reporting beneficial spatial memeory deficits and loss of thalamo cortical connectivity resulting stroke in mice. The experimental design and methodology appear sound, however the statistical analysis is not justified and the quality of the figures is so poor one could not determine if the conclusions are supported by the results.

Reviewer 1 has acknowledged that the statistical analyses used in this manuscript was appropriate given the inability to compare hydrogel treatments in sham animals.

We have also improved the quality of the figures and saved these in higher dpi.

Other more minor points:

The exclusive use of males is not justified other. Women do get stroke.

Authors agree with this comment, but must acknowledge the size of this study means to add in the additional factor of testing females dramatically increases the experimental cohort to an extend it is infeasible to test as a single cohort. Future research should indeed investigate the use of the current drug combinations in females to evaluate the translatability of this intervention.

The occasional sentence is uninterpretable.

These have been amended as per the changes recommended by Reviewer 1.

Comments on these issues are detailed above in order of appearance in the MS.

Intro:

“Indeed, GAGs have been  observed to act as medium-affinity co-receptors and bind to GFs ( ..) to their respective receptors”. What does this sentence mean?

Changed for clarity -  “Indeed, GAGs themselves have been observed to act as medium-affinity co-receptors to help bind GFs (e.g. brain-derived neurotrophic factor, BDNF) to their respective receptors

Methods

Michael’s reaction and similar needs to be referenced if not described in detail.

Michael reaction is a standard chemical reaction and doesn’t require a reference here.

P3 line 109 “compound A and compound, ??? demonstrating…Presumably conpound G?

Changed to “… compound A and compound G…”

Animals:

Animals need to be described first,  in terms of strain, sex, age, or weight. This is also a good place to mention the total number of animals included in all of the studies described.

The description of the animals has been moved to follow the first mention of the mice (“Prior to behavioural testing, mice (2-3 months old, male, C57BL/6J) were housed…”).

A supplementary table has also been added to the manuscript to give a summary of total sample sizes used within each treatment group and were numbers were lost for each analysis (e.g. CTB, GFAP). This is also referred to in text, “A summary of all group sizes including exclusions for downstream analyses (e.g. behaviour, CTB tracing, GFAP quantification) can be found in Supplementary Table 1.

Line 122 “All aged animals randomly assigned to a treatment group”. What does this mean?

The means of randomisation was clarified in the statement, “with use of an online randomization program (http://www.randomization.com)”. The term “Aged” was a typo and has been removed from that sentence.

“A single prolonged dose” is not a good choice of words…

Changed to “… a single, prolonged dose…”

“Loaded pumps when placed port side facing away from the incision before the incision site was closed.”  Please rephrase

Authors have now acknowledged prior to this statement that there is a ‘loading site’ on the minipumps (“…drug solution was slowly loaded into the loading sites of the minipumps…”). Remaining paragraph was also altered as follows, “Loaded-minipumps were inserted (with the loading port facing away from the incision), via a small (<1cm) incision made between the shoulder blades of the animals, into a subcutaneous pocket made using blunt forceps under the skin. Loaded pumps when placed port side facing away from the incision before the incision site was closed.”

“IgG-Fc_(5μg/mL)_or_rhBDNF_(0.167μg_/μL)_” – are the concetration units correct? If yes, why the big difference in concentration between active and control? 

The concentrations reported are the same as what we have used previously.  We have corrected the concentration for BDNF to read as 167ug/ml to have the same units as IgG-Fc.

What is temgesic?

Clarified in text – “Temgesic® (Buprenorphine hydrochloride)…”

FIJI and Image J are the same software (Fiji is Image J…) but the software is referred to as “image J” in one place and” FIJI Image J”  in another . This is confusing.

Have changed to “FIJI” (NIH, USA) for clarity

It is not clear how infarct volume was measured, considering the infarcted area is not enclosed by tissue ( as would be the case with a striatal infarct). Under these circumstances, the dorsal ”borders “ of the infarct can not be seen and the volume of the infarct can not be measured directly. There are different solutions to this problem but it it’s not clear what the authors have done.

Amended in the manuscript: “Dorsal borders of the infarct were first estimated by tracing the remaining cerebral tissue back to the midline (ensuring a symmetrical curvature across left and right hemispheres), before infarct areas (mm2) were calculated. Infarct volume were then quantified as follows:”

The rationale for the described  statistical analysis is not clear.

See comment in opening statement.

Results

Fig 3. Is very confusing, stacking the red and blue bars on top of each other to denote right and left hemisphere is very confusing. This figure needs to be redrawn.

We kindly disagree with the reviewer on this point and have published previously using this figure format, which has been received with positive feedback from many other reviewers. We believe it is important to show that the infarct volume in both the left and right hemispheres was equal across all treatment groups, as asymmetrical differences in PFC lesions has been shown to alter cognitive impairment. For this reason we have chosen to leave this figure unchanged.

It is not clear what is meant by “multiple comparisons”, do the authors mean posthoc comparisons?

Multiple comparisons” has been rewritten as “post hoc comparisons” throughout the manuscript.

The results of the  statistical analysis are not detailed enough – for both 2 way and one way ANOVA the main effects (F, p)need to be reported first, prior to results of posthoc anslaysis.  In the absence of a main effect or interaction, posthoc comparisons are not justified.

All results sections already follow this format, first reporting any overall group effects (one-way = treatment effect; two-way = stroke, treatment, interaction); F and p values included followed by reporting any post hoc / multiple comparisons where appropriate (as p values).

“compared to animals treated with vehicle-treated stroke animals”-needs fixing

Amended to “compared to vehicle-treated stroke animals

 Figure 6 is of such bad quality /small size it is uninterpretable. Same for figure 7, 8 and 9.

We apologise for this issue as the figures looked fine on our computers.  We have now increased the size of the photomicrographs and saved the figures in higher dpi to address this issue. 
